# SnCl_4_ Promoted Efficient Cleavage of Acetal/Ketal Groups with the Assistance of Water in CH_2_Cl_2_

**DOI:** 10.3390/molecules27238258

**Published:** 2022-11-26

**Authors:** Tao Luo, Tian-Tian Xu, Yang-Fan Guo, Hai Dong

**Affiliations:** Key Laboratory of Material Chemistry for Energy Conversion and Storage, Ministry of Education, Hubei Key Laboratory of Material Chemistry and Service Failure, School of Chemistry & Chemical Engineering, Huazhong University of Science & Technology, Luoyu Road 1037, Wuhan 430074, China

**Keywords:** glycosides, SnCl_4_, 4,6-arylidene, deacetalization, selectivity

## Abstract

Acetalization and deacetalation are a pair of routine manipulations to protect and deprotect the 4- and 6-hydroxyl groups of glycosides in the synthesis of glycosyl building blocks. In this study, we found that treatment of SnCl_4_ with various carbohydrates containing acetal/ketal groups with the assistance of water in CH_2_Cl_2_ led to deacetalization/deketalization products in almost quantitative yields. In addition, for substrates containing both acetal/ketal and *p*-methoxylbenzyl groups, we also found that the *p*-methoxylbenzyl group was selectively cleaved by the use of a catalytic amount of SnCl_4_, while the acetal/ketal groups remained. Furthermore, based on this, 4,6-benzylidene glycosides can be conveniently converted to 4,6-OAc or 4-OH, 6-OAc glycosides.

## 1. Introduction

Orthogonally protected glycosyl building blocks play key roles in the synthesis of oligosaccharides, whose preparation usually requires multiple steps of selective protection and deprotection [1,2,3,4,5,6,7]. Acetalation is a routine manipulation for protecting the 4- and 6-hydroxyl groups of glycosides in the synthesis of glycosyl building blocks [8,9,10,11,12,13,14,15,16,17], and thus methods for removing 4,6-arylidene acetals were extensively reported [18,19,20,21,22,23,24,25,26,27,28,29,30,31,32,33]. Acidic hydrolysis of the 4,6-arylidene group is the most commonly used method, including the use of AcOH [18,19], CF_3_COOH [18,19], VO(OTf)_2_ [20], Er(OTf)_3_ [21], SnCl_2_ [22], NaHSO_4_ [23], I_2_ [24] and so on (Entries 1–7 in Table 1). Improved methods included the combined use of Lewis acids and dithiols (Entries 8–9) [25,26], and the use of silica gel supported acids (Entries 10–15) [27,28,29,30,31,32]. In addition, an improved method for deprotection of the 4,6-arylidene group under hydrogenation conditions was to use Et_3_SiH instead of H_2_ (Entry 16) [33]. These methods each have their own advantages and disadvantages, and the disadvantages usually include relatively harsh conditions, long reaction times, incompatibility with many functional groups and the formation of unwanted byproducts. SnCl_4_, as a Lewis acid, was used in the selective removal of benzyl groups in carbohydrate synthesis, but exhibited relatively low reactivity [34]. Considering the difference in the stability of benzyl and benzylidene under acidic conditions, we then attempted to use SnCl_4_ to achieve rapidly and highly selective removal of acetal and ketal groups. In this study, we found that treatment of carbohydrates containing acetal/ketal groups with SnCl_4_ ith the assistance of water in CH_2_Cl_2_ (DCM) led to deacetalization/deketalization products in almost quantitative yields. For substrates containing both acetal/ketal and PMB groups, we also found that the PMB group was selectively cleaved by the use of a catalytic amount of SnCl_4_ in DCM, while the acetal/ketal groups remained (Figure 1a). Furthermore, based on SnCl_4_-promoted deacetalization, 4,6-benzylidene glycosides can be conveniently converted to 4,6-OAc glycosides (Figure 1b) or 4-OH, 6-OAc glycosides (Figure 1c).

## 2. Results

We first evaluated the potential of SnCl_4_ to remove the 4,6-*O*-benzylidene group of glycosides using methyl 2,3-di-*O*-benzyl-4,6-*O*-benzylidene-α-D-glucopyranoside **1a** as a model compound. Thus, **1a** was allowed to react with SnCl_4_ in DCM at rt (Table 2). Interestingly, as the used amount of SnCl_4_ was gradually increased from 0.2 equiv to 2.5 equiv, the yield of the deacetalation product **2** increased from 19% to a nearly quantitative yield (Entries 1–3). These results seem to support an equilibrium reaction. The reaction mechanism is proposed in Figure 1a, where the coordination of SnCl_4_ to the 4,6-oxygen atoms of **1a** leads to the cleavage of the benzylidene group, and the formation of the intermediate **M** and dichlorotoluene. However, we failed when we tried to capture dichlorotoluene by an NMR experiment to support this mechanism (Appendix A). The fact that benzaldehyde was captured instead of dicholorotoluene supports the mechanism shown in Figure 1b, where trace amounts of water play an indispensable role in the cleavage of the benzylidene group. The mechanism also explained why it is not feasible to use a catalytic amount of SnCl_4_ in the reaction. When the solvent used in the reaction was changed from DCM to the more polar methanol and acetonitrile, the yields of **2** were greatly reduced (Entry 4). The reason must be due to the competitive coordination of SnCl_4_ with polar solvents.

As can be seen from the NMR spectrum (Appendix A), the reaction was terminated when the trace amounts of water in the *d*-choloroform was consumed. We then tried using water to assist this reaction (Entries 5–8). As can be seen, optimal conditions were to use 1.2–1.5 equiv of SnCl_4_ and 1.0–1.5 equiv of water; reaction at rt for 10 min under these conditions led to **2** in a nearly quantitative yield (Entry 8). Similar results for Entries 5 and 6 indicate that the simultaneous addition of SnCl_4_ and water had no adverse effect on the yield of **2**. However, methanol used as a hydrogen source instead of water in the reaction proved to be ineffective (Entry 9). The use of HCl, SnCl_2_, FeCl_3_, CuCl_2_ and Cu(OTf )_2_ instead of SnCl_4_ in the reaction resulted in varying degrees of reduced yields of **2** (Entries 10–12). We also envisaged that the reaction of AcCl/Ac_2_O with **M** might produce selectively acetylated products. Therefore, AcCl/Ac_2_O instead of water was added to the reaction. However, the addition of AcCl led to the formation of a complex mixture (Entry 13), and the addition of Ac_2_O led to the formation of 4,6-OAc product **2a** as the main product (Entry 14), indicating poor selectiveacetylations.

With the optimized conditions in hand, 4,6-*O*-arylidene glycosides **1b**, **1c**, **3**, **5**, **7**, **11**, **13**, **15**, **17** and **19** were further evaluated in the reaction (Entries 1–6 in Table 3). It can be seen that deacetalized products **2**, **4**, **6**, **8**, **10**, **12**, **14**, **16**, **18** and **20** were obtained in 89–98% yields after treating the substrates with 1.5 equiv of SnCl_4_ in the presence of 1.0 equiv of water in DCM for 10 min at rt. The results indicate that this method was compatible with various configurations of glycosides and functional groups. In particular, unlike the reported effect of SnCl_4_ on 1-STol/SBn glycosides [35], SnCl_4_ did not cause cleavage or configurational isomerization of the 1-STol/SBn group for 1-STol/SBn glycoside substrates (Entry 4). Encouraged by these results, we further tested removal of the isopropylidene ketal protecting group under these conditions. Similarly, the deketalized products **22**, **24** and **26** were obtained in 94–97% yields from **21**, **23** and **25** after only 10 min of reaction at rt (Entries 7 and 8). Especially, for the 1,2,5,6-diisopropylidene ketal-protected furanose substrates **23** and **25**, the 5,6-isopropylidene ketal were preferentially removed to obtain the 1,2-isopropylidene ketal-protected furanose products **24** and **26** in excellent yields by this method (Entry 8).

We also noticed that the 4-methoxylbenzyl (PMB) protecting group could be deprotected in the presence of catalytic amounts of SnCl_4_ [36]. The catalytic mechanism involved the formation of the desired alcohol, the release of a PMB cation, and the subsequent formation of a lipophilic side product through the Friedel–Crafts alkylation of the PMB cation with another PMB ether [37,38]. Indeed, after treatment of methyl 2,4,6-tri-*O*-acetyl-3-*O*-PMB-galactoside/mannoside **27**/**29** with 0.2 equiv of SnCl_4_ in DCM at rt for 10 min, the PMB-removed products **28**/**30** were obtained in 92/98% yield (Entries 9 and 10). Since the reactivity for removing PMB is much higher than that for removing acetal/ketal by SnCl_4_, we guessed that PMB should be preferentially removed from substrates containing both PMB and acetal/ketal in the presence of catalytic amounts of SnCl_4_. Therefore, four substrates **31**, **33**, **35** and **37** containing both PMB and acetal/ketal were treated with 0.2–0.5 equiv of SnCl_4_ in DCM at rt for 5 min, leading to the selective PMB-removed products **32**, **34**, **36** and **38** in 80–85% yields (Figure 2).

In the control experiment shown for Entry 14 in Table 2, the 4,6-benzylidene acetal was conveniently converted to 4,6-OAc for **1a** when acetic anhydride was used instead of H_2_O during the SnCl_4_-promoted deacetalation; in addition, a 78% yield of **2a** was obtained when acetic anhydride was directly added dropwise to the reaction mixture. Further experiments indicate that the yield of **2a** increased to 91% when a solution of acetic anhydride in dry acetonitrile was added dropwise to the reaction mixture (Figure 3). Using this method, 4,6-OAc glycosides **41** (82%), **42** (85%) and **43** (88%) were obtained in high yields from 4,6-benzylidene glycosides **15**, **39** and **40** (Figure 3). As seen in Table 2, the SnCl_4_-mediated deacetalation method led to deprotected diol products in almost quantitative yields. We thus envisioned a directly selective acetylation of 6-OH after deprotection of 4,6-benzylidene glycosides without purification of the 4,6-OH glycoside products. After the SnCl_4_-promoted deacetalation of 4,6-O-benzylidene glycosides **1a**, **15**, **39** and **40** was completed, the reaction mixture was dissolved in dichloromethane and extracted using a saturated sodium bicarbonate solution and a saturated sodium potassium tartrate solution. The concentrated crude products were then dissolved in dry acetonitrile, followed by the addition of 1.1 equiv of Ac_2_O and 0.2 equiv of DIPEA [39]. The reaction proceeded at 40 °C for 12 h, resulting in 6-OAc products **44**, **45**, **46** and **47** in 70%, 64%, 69% and 65% yields, respectively (Figure 3).

## 3. Conclusions

In this study, it was found that acetal and ketal protective groups could be efficiently removed in the presence of SnCl_4_ for orthogonally protected carbohydrate substrates. The reaction can be completed in DCM within 10 min at room temperature, and a small amount of water has an obvious promoting effect on the reaction. It was also found that the PMB could be preferentially removed from the substrates containing both acetal/ketal and PMB by a catalytic amount of SnCl_4_. Based on SnCl_4_-promoted deacetalation, 4,6-benzylidene glycosides can be conveniently converted to 4,6-OAc glycosides and 4-OH, 6-OAc glycosides. These methods provide efficient approaches to synthesizing orthogonally protected carbohydrate building blocks.

## 4. Materials and Methods

**General Methods**. All chemicals were purchased as reagent grade and used without further purification. The solvents were purified before use and CH_3_CN was distilled from CaH_2_. Chemical reactions were monitored by thin-layer chromatography using precoated silica gel 60 (0.25 mm thickness) plates. Flash column chromatography was performed on silica gel 60 (SDS 0.040–0.063 mm). Spots were visualized by UV light (254 nm) then by charring with a solution of H_2_SO_4_ (5%) in ethanol. ^1^H NMR spectra were recorded by 400 MHz or 600 MHz (^1^H) and 100 MHz (^13^C) at 298 K in CDCl_3_ using the residual signals from CDCl_3_ (^1^H: δ = 7.26 ppm; ^13^C: δ = 77.16 ppm) or CD_3_OD (^1^H: δ = 3.31 ppm) as the internal standard. ^1^H peak assignments were made by first order analysis of the spectra, supported by standard ^1^H–^1^H correlation spectroscopy (COSY). High-resolution mass spectra (HRMS) were obtained by TOF detection. Optical rotations were measured on an SGW-1 automatic polarimeter with [α]_D_ values reported in degrees; concentration (c) is in g/100 mL.

**General procedure A for SnCl_4_-mediated deacetalization**. SnCl_4_ (1.5 equiv) and H_2_O (1.0 equiv) were added to a solution of a carbohydrate substrate containing acetal/ketal in DCM (1 mL). The mixture was stirred at rt for 10 min and then poured onto a cold saturated NaHCO_3_ solution. The organic phase was separated and the aqueous phase was extracted with dichloromethane (3 × 10 mL). The combined organic phase was washed with saturated sodium potassium tartrate solution (1 × 15 mL), dried with anhydrous MgSO_4_, and concentrated in vacuo. The residue was purified by silica gel flash chromatography.

**General procedure B for SnCl_4_-mediated removal of PMB**. SnCl_4_ (0.2–0.5 equiv) was added to a solution of a carbohydrate substrate containing PMB in DCM (1 mL). The mixture was stirred at rt for 5 min and then poured onto a cold saturated NaHCO_3_ solution. The organic phase was separated and the aqueous phase was extracted with dichloromethane (3 × 10 mL). The combined organic phase was washed with the saturated NaHCO_3_ solution (1 × 15 mL), dried with anhydrous MgSO_4_, and concentrated in vacuo. The residue was purified by silica gel flash chromatography.

**General procedure C for converting of 4,6-benzylidene to 4,6-OAc for glycosides**. SnCl_4_ (1.5 equiv) and a solution of acetic anhydride (2.0 equiv), added dropwise in dry acetonitrile (0.5 mL), were added to a solution of a 4,6-benzylidene glycoside in DCM (1 mL). The mixture was stirred at rt for 10 min and then poured onto a cold saturated NaHCO_3_ solution. The organic phase was separated and the aqueous phase was extracted with dichloromethane (3 × 10 mL). The combined organic phase was washed with the saturated NaHCO_3_ solution (1 × 15 mL), dried with anhydrous MgSO_4_, and concentrated in vacuo. The residue was purified by silica gel flash chromatography.

**General procedure D for converting of 4,6-benzylidene to 4-OH, 6-OAc for glycosides**. SnCl_4_ (1.5 equiv) and H_2_O (1.0 equiv) were added to a solution of a carbohydrate substrate containing acetal/ketal in DCM (1 mL). The mixture was stirred at rt for 10 min and then poured onto cold saturated a NaHCO_3_ solution. The organic phase was separated and the aqueous phase was extracted with dichloromethane (3 × 10 mL). The combined organic phase was washed with saturated sodium potassium tartrate solution (1 × 15 mL), dried with anhydrous MgSO_4_, and concentrated in vacuo. The residue was allowed to react with acetic anhydride (1.1 equiv) in the presence of DIPEA (0.2 equiv) in dry acetonitrile (1 mL) at 40 °C for 12 h. After cooling and evaporation of the solvent, the reaction mixture was directly purified by flash column chromatography.

*Methyl 2,3-di-O-benzyl-α-D-glucopyranoside* (**2**) [26]. Following general process A, starting from **1a** (100 mg, 0.22 mmol), purification by column chromatography (petroleum ether/ethyl acetate 1/1.5) afforded **2** as a white solid (77 mg, 95%). ^1^H NMR (600 MHz, CDCl_3_) δ 7.40–7.28 (m, 10H), 5.03 (d, *J* = 11.5 Hz, 1H), 4.77 (d, *J* = 12.0 Hz, 1H), 4.70 (d, *J* = 11.5 Hz, 1H), 4.66 (d, *J* = 12.0 Hz, 1H), 4.60 (d, *J* = 3.5 Hz, 1H), 3.83–3.76 (m, 2H), 3.74 (dd, *J* = 11.5, 4.6 Hz, 1H), 3.62 (dt, *J* = 8.4, 4.1 Hz, 1H), 3.55–3.47 (m, 2H), 3.38 (s, 3H).

*Methyl 2,3-di-O-methyl-α-D-glucopyranoside* (**4**) [40]. Following general process A, starting from **3** (72 mg, 0.23 mmol), purification by column chromatography (petroleum ether/ethyl acetate 1/1.5) afforded **4** as a white solid (46.7 mg, 91%). ^1^H NMR (400 MHz, CDCl_3_) δ 4.85 (d, *J* = 3.5 Hz, 1H), 3.92–3.74 (m, 2H), 3.64 (s, 4H), 3.53–3.46 (m, 5H), 3.44 (s, 3H), 3.28–3.18 (m, 1H).

*Methyl 2,3-di-O-acetyl-α-D-glucopyranoside* (**6**) [26]. Following general process A, starting from **5** (41 mg, 0.11 mmol), purification by column chromatography (petroleum ether/ethyl acetate 1/1.5) afforded **6** as a colorless oil (30.5 mg, 98%). ^1^H NMR (400 MHz, CDCl_3_) δ 5.34–5.25 (m, 1H), 4.91 (d, *J* = 3.6 Hz, 1H), 4.83 (dd, *J* = 10.1, 3.6 Hz, 1H), 3.95– 3.82 (m, 2H), 3.76–3.65 (m, 2H), 3.40 (s, 3H), 2.11 (s, 3H), 2.09 (s, 3H).

*Methyl 2,3-di-O-pivaloyl-α-D-glucopyranoside* (**8**) [41]. Following general process A, starting from **7** (28.2 mg, 0.063 mmol), purification by column chromatography (petroleum ether/ethyl acetate 1/1.5) afforded **8** as a white solid (22 mg, 97%). ^1^H NMR (400 MHz, CD_3_OD) δ 5.46–5.30 (m, 1H), 4.73 (dd, *J* = 10.1, 3.6 Hz, 1H), 3.85 (dd, *J* = 11.9, 2.2 Hz, 1H), 3.74 (dd, *J* = 11.9, 5.1 Hz, 1H), 3.66 (ddd, *J* = 10.1, 5.1, 2.2 Hz, 1H), 3.58 (t, *J* = 9.5 Hz, 1H), 3.42 (s, 3H), 3.36–3.30 (m, 1H), 1.21 (s, 9H), 1.18 (s, 9H).

*Methyl 2,3-di-O-benzoyl-β-D-glucopyranoside* (**10**) [26]. Following general process A, starting from **9** (64 mg, 0.13 mmol), purification by column chromatography (petroleum ether/ethyl acetate 1/1.5) afforded **10** as a colorless oil (49.8 mg, 95%). ^1^H NMR (400 MHz, CDCl_3_) δ 8.02–7.89 (m, 4H), 7.51 (t, *J* = 7.8 Hz, 2H), 7.37 (td, *J* = 7.8, 3.6 Hz, 4H), 5.49–5.28 (m, 2H), 4.74–4.54 (m, 1H), 4.04–3.90 (m, 3H), 3.60 (dt, *J* = 9.2, 4.0 Hz, 1H), 3.53 (s, 3H).

*Benzyl 2,3-di-O-acetyl-1-thio-β-D-glucopyranoside* (**12**). Following general process A, starting from **11** (25 mg, 0.055 mmol), purification by column chromatography (petroleum ether/ethyl acetate 1/1.5) afforded **12** as a colorless oil (18.8 mg, 93%). [α]_D_^12^ = −88 (c 0.2, CH_2_Cl_2_), ^1^H NMR (400 MHz, CDCl_3_) δ 7.42–7.23 (m, 5H), 5.07–4.91 (m, 2H), 4.39 (d, *J* = 8.0 Hz 1H), 3.95–3.87 (m, 3H), 3.80–3.64 (m, 2H), 3.39–3.17 (m, 2H), 2.34–2.18 (b, 1H), 2.08 (s, 3H), 2.03 (s, 3H). ^13^C NMR (100 MHz, CDCl_3_) δ 171.58, 169.68, 137.45, 128.95, 128.61, 128.56, 127.38, 82.63, 79.65, 69.69, 69.28, 62.12, 34.40, 20.86, 20.74. HRMS (ESI-TOF) m/z [M + Na]^+^ calcd for [C_17_H_22_O_7_SNa]^+^: 393.0984; found: 393.0964.

*4-Methylphenyl 2,3-di-O-acetyl-1-thio-D-glucopyranoside* (**14**) [25]. Following general process A, starting from **13** (26 mg, 0.06 mmol), purification by column chromatography (petroleum ether/ethyl acetate 1/1.5) afforded **14** as a colorless oil (19.4 mg, 94%). ^1^H NMR (400 MHz, CDCl_3_) δ 7.41–7.35 (m, 2H), 7.15 (d, *J* = 7.9 Hz, 2H), 5.05 (t, *J* = 9.3 Hz, 1H), 4.92 (t, *J* = 9.6 Hz, 1H), 4.69 (d, *J* = 10.0 Hz, 1H), 4.02–3.89 (m, 1H), 3.83 (d, *J* = 12.2 Hz, 1H), 3.73 (t, *J* = 9.6 Hz, 1H), 3.53–3.42 (m, 1H), 2.36 (s, 3H), 2.12 (s, 3H), 2.10 (s, 3H).

*Methyl 2,3-di-O-benzyl-α-D-galactopyranoside* (**16**) [20]. Following general process A, starting from **15** (53 mg, 0.11 mmol), purification by column chromatography (petroleum ether/ethyl acetate 1/1.5) afforded **16** as a colorless oil (38 mg, 89%). ^1^H NMR (600 MHz, CDCl_3_) δ 7.42–7.26 (m, 10H), 4.81 (d, *J* = 11.8 Hz, 2H), 4.70 (dd, *J* = 7.4, 4.2 Hz, 2H), 4.67 (d, *J* = 12.1 Hz, 1H), 4.05 (dd, *J* = 2.9, 1.4 Hz, 1H), 3.93–3.83 (m, 3H), 3.79–3.73 (m, 2H), 3.38 (s, 3H).

*Methyl 2,3-di-O-benzyl-β-D-galactopyranoside* (**18**) [42]. Following general process A, starting from **17** (48 mg, 0.1 mmol), purification by column chromatography (petroleum ether/ethyl acetate 1/1.5) afforded **18** as a white solid (38.8 mg, 90%). ^1^H NMR (400 MHz, CDCl_3_) δ 7.42–7.27 (m, 10H), 4.90 (d, *J* = 11.0 Hz, 1H), 4.77–4.68 (m, 3H), 4.30 (d, *J* = 7.7 Hz, 1H), 4.04–3.93 (m, 2H), 3.83 (dd, *J* = 11.7, 4.5 Hz, 1H), 3.67–3.61 (m, 1H), 3.58 (s, 3H), 3.53–3.46 (m, 2H), 2.66 (s, 1H), 2.21 (t, *J* = 7.6 Hz, 1H).

*Methyl 2,3-di-O-acetyl-α-D-mannopyranoside* (**20**) [43]. Following general process A, starting from **19** (27 mg, 0.07 mmol), purification by column chromatography (petroleum ether/ethyl acetate 1/1.5) afforded **20** as a white solid (19.5 mg, 95%). ^1^H NMR (400 MHz, CDCl3) δ 5.22 (dd, *J* = 3.5, 1.7 Hz, 1H), 5.17 (dd, *J* = 9.9, 3.4 Hz, 1H), 4.67 (d, *J* = 1.7 Hz, 1H), 3.98 (t, *J* = 9.8 Hz, 1H), 3.89 (dd, *J* = 4.0, 2.1 Hz, 2H), 3.70 (dt, *J* = 9.8, 3.9 Hz, 1H), 3.39 (s, 3H), 2.69 (s, 1H), 2.12 (s, 3H), 2.07 (s, 3H), 1.79 (s, 1H).

*Methyl 2,6-di-O-benzyl-α-D-galactopyranoside* (**22**) [44]. Following general process A, starting from **21** (64 mg, 0.15 mmol), purification by column chromatography (petroleum ether/ethyl acetate 1/1.5) afforded **22** as a colorless oil (56 mg, 97%). ^1^H NMR (400 MHz, CDCl_3_) δ 7.43–7.22 (m, 10H), 4.70 (d, *J* = 3.7 Hz, 1H), 4.70–4.61 (m, 2H), 4.57 (d, *J* = 1.9 Hz, 2H), 4.04 (d, *J* = 3.4 Hz, 1H), 3.96 (dd, *J* = 9.8, 3.4 Hz, 1H), 3.94–3.86 (m, 1H), 3.76–3.68 (m, 3H), 3.35 (s, 3H).

*3-O-benzoyl-1,2-O-isopropylidene-α-D-glucofuranose* (**24**) [45]. Following general process A, starting from **23** (78.7 mg, 0.22 mmol), purification by column chromatography (petroleum ether/ethyl acetate 1/1.5) afforded **24** as a colorless oil (66.1 mg, 94%). ^1^H NMR (400 MHz, CDCl_3_) δ 8.14–7.98 (m, 2H), 7.67–7.57 (m, 1H), 7.47 (t, *J* = 7.7 Hz, 2H), 6.01 (d, *J* = 3.7 Hz, 1H), 5.53 (d, *J* = 2.6 Hz, 1H), 4.73 (d, *J* = 3.7 Hz, 1H), 4.30 (dd, *J* = 8.6, 2.6 Hz, 1H), 3.87 (d, *J* = 8.1 Hz, 1H), 3.75 (d, *J* = 8.3 Hz, 2H), 3.19 (s, 1H), 2.29–2.13 (m, 1H), 1.56 (s, 3H), 1.34 (s, 3H).

*3-O-Benzyl-1,2-O-isopropylidene-α-D-glucofuranose* (**26**) [26]. Following general process A, starting from **25** (139 mg, 0.4 mmol), purification by column chromatography (petroleum ether/ethyl acetate 1/1.5) afforded **26** as a colorless oil (112 mg, 91%). ^1^H NMR (400 MHz, CDCl_3_) δ 7.44–7.29 (m, 5H), 5.96 (d, *J* = 3.8 Hz, 1H), 4.75 (d, *J* = 11.8 Hz, 1H), 4.65 (d, *J* = 3.8 Hz, 1H), 4.58 (d, *J* = 11.7 Hz, 1H), 4.14–4.09 (m, 2H), 4.05–4.00 (m, 1H), 3.82 (dd, *J* = 11.5, 3.4 Hz, 1H), 3.71 (dd, *J* = 11.5, 5.4 Hz, 1H), 2.70 (s, 1H), 1.91 (s, 1H), 1.50 (s, 3H), 1.34 (s, 3H).

*Methyl 2,4,6-tri-O-acetyl-α-D-galactopyranoside* (**28**) [46]. Following general process B, starting from **27** (45 mg, 0.1 mmol), purification by column chromatography (petroleum ether/ethyl acetate 2/1) afforded **28** as a white solid (30 mg, 92%). ^1^H NMR (400 MHz, CDCl_3_) δ 5.35 (dd, *J* = 3.7, 1.2 Hz, 1H), 4.98 (dd, *J* = 10.1, 8.0 Hz, 1H), 4.38 (d, *J* = 7.9 Hz, 1H), 4.22–4.17 (m, 2H), 3.91–3.84 (m, 1H), 3.54 (s, 3H), 2.58 (d, *J* = 6.3 Hz, 1H), 2.20 (s, 3H), 2.16 (s, 3H), 2.09 (s, 3H).

*Methyl 2,4,6-tri-O-acetyl-α-D-mannopyranoside* (**30**) [43]. Following general process B, starting from **29** (88 mg, 0.2 mmol), purification by column chromatography (petroleum ether/ethyl acetate 2/1) afforded **30** as a colorless oil (53 mg, 98%). ^1^H NMR (400 MHz, CDCl_3_) δ 5.15–5.00 (m, 2H), 4.77 (d, *J* = 1.6 Hz, 1H), 4.31 (dd, *J* = 12.2, 5.4 Hz, 1H), 4.14 (dd, *J* = 12.1, 2.3 Hz, 1H), 4.10–4.03 (m, 1H), 3.94–3.89 (m, 1H), 3.39 (s, 3H), 2.33–2.26 (m, 1H), 2.17 (s, 3H), 2.13 (s, 3H), 2.11 (s, 3H).

*Methyl 3-O-benzyl-4,6-di-O-benzylidene-α-D-glucopyranoside* (**32**) [5]. Following general process B, starting from **31** (138 mg, 0.28 mmol), purification by column chromatography (petroleum ether/ethyl acetate 3/1) afforded **32** as a colorless oil (85.4 mg, 82%). ^1^H NMR (400 MHz, CDCl_3_) δ 7.57–7.43 (m, 2H), 7.44–7.23 (m, 9H), 5.58 (s, 1H), 4.97 (d, *J* = 11.6 Hz, 1H), 4.89–4.70 (m, 2H), 4.30 (dd, *J* = 9.8, 4.3 Hz, 1H), 3.91–3.67 (m, 4H), 3.65 (t, *J* = 9.1 Hz, 1H), 3.46 (s, 3H).

*Methyl 3-O-(tert-butyl-dimethylsilyl)-4,6-di-O-benzylidene-α-D-glucopyranoside* (**34**) [47]. Following general process B, starting from **33** (49 mg, 0.1 mmol), purification by column chromatography (petroleum ether/ethyl acetate 3/1) afforded **34** as a colorless oil (30 mg, 80%). ^1^H NMR (400 MHz, CDCl_3_) δ 7.55–7.44 (m, 2H), 7.36 (dd, *J* = 5.2, 2.0 Hz, 3H), 5.50 (s, 1H), 4.81 (d, *J* = 3.9 Hz, 1H), 4.27 (dd, *J* = 9.4, 4.1 Hz, 1H), 3.90 (t, *J* = 9.0 Hz, 1H), 3.81–3.71 (m, 3H), 3.59 (d, *J* = 3.9 Hz, 1H), 3.45 (s, 3H), 2.13 (d, *J* = 7.8 Hz, 1H), 0.87 (s, 9H), 0.10 (s, 3H), 0.02 (s, 3H).

*1,2:5,6-Di-O-isopropylidene-α-D-glucofuranose* (**36**) [48]. Following general process B, starting from **35** (179 mg, 0.47 mmol), purification by column chromatography (petroleum ether/ethyl acetate 3/1) afforded **36** as a white solid (104 mg, 85%). ^1^H NMR (400 MHz, CDCl_3_) δ 5.95 (d, *J* = 3.6 Hz, 1H), 4.54 (d, *J* = 3.6 Hz, 1H), 4.37–4.33 (m, 2H), 4.17 (dd, *J* = 8.6, 6.2 Hz, 1H), 4.07 (dd, *J* = 7.6, 2.8 Hz, 1H), 3.99 (dd, *J* = 8.7, 5.4 Hz, 1H), 2.58 (d, *J* = 3.5 Hz, 1H), 1.50 (s, 3H), 1.45 (s, 3H), 1.37 (s, 3H), 1.32 (s, 3H).

*1,2:3,4-Di-O-isopropylidene-α-D-galactopyranose* (**38**) [49]. Following general process B, starting from **37** (65 mg, 0.17 mmol), purification by column chromatography (petroleum ether/ethyl acetate 3/1) afforded **38** as a colorless oil (37.1 mg, 84%). ^1^H NMR (400 MHz, CDCl_3_) δ 5.57 (d, *J* = 5.0 Hz, 1H), 4.62 (dd, *J* = 8.0, 2.4 Hz, 1H), 4.34 (dd, *J* = 5.1, 2.4 Hz, 1H), 4.32–4.23 (m, 1H), 3.87 (d, *J* = 9.4 Hz, 2H), 3.82–3.68 (m, 1H), 1.54 (s, 3H), 1.46 (s, 3H), 1.34 (s, 6H).

*Methyl 2,3-di-O-benzyl-4,6-di-O-acetyl-α-D-glucopyranoside* (**2a**) [29]. Following general process C, starting from **1a** (100 mg, 0.22 mmol), purification by column chromatography (petroleum ether/ethyl acetate 3/1) afforded **2a** as a white solid (90 mg, 91%). ^1^H NMR (400 MHz, CDCl_3_) δ 7.42–7.28 (m, 10H), 4.99 (dd, *J* = 10.3, 9.3 Hz, 1H), 4.89 (d, *J* = 11.6 Hz, 1H), 4.81 (d, *J* = 12.1 Hz, 1H), 4.65 (d, *J* = 12.1 Hz, 2H), 4.59 (d, *J* = 3.6 Hz, 1H), 4.21 (dd, *J* = 12.1, 4.9 Hz, 1H), 4.00 (dd, *J* = 12.1, 2.3 Hz, 1H), 3.92 (t, *J* = 9.4 Hz, 1H), 3.88–3.82 (m, 1H), 3.59 (dd, *J* = 9.6, 3.5 Hz, 1H), 3.39 (s, 3H), 2.06 (s, 3H), 1.91 (s, 3H).

*Methyl 2,3-di-O-benzyl-4,6-di-O-acetyl-α-D-galactopyranoside* (**41**) [29]. Following general process C, starting from **15** (80 mg, 0.17 mmol), purification by column chromatography (petroleum ether/ethyl acetate 3/1) afforded **41** as a colorless oil (65 mg, 82%). ^1^H NMR (400 MHz, CDCl_3_) δ 7.43–7.22 (m, 10H), 5.54 (d, *J* = 3.4 Hz, 1H), 4.85 (d, *J* = 12.1 Hz, 1H), 4.74 (d, *J* = 11.2 Hz, 1H), 4.69 (d, *J* = 3.6 Hz, 1H), 4.65 (d, *J* = 12.1 Hz, 1H), 4.57 (d, *J* = 11.1 Hz, 1H), 4.14–4.05 (m, 3H), 3.97 (dd, *J* = 10.0, 3.5 Hz, 1H), 3.78 (dd, *J* = 10.0, 3.7 Hz, 1H), 3.39 (s, 3H), 2.13 (s, 3H), 2.07 (s, 3H).

*Methyl 2,3-di-O-benzyl-4,6-di-O-acetyl-α-D-mannopyranoside* (**42**) [50]. Following general process C, starting from **39** (180 mg, 0.4 mmol), purification by column chromatography (petroleum ether/ethyl acetate 3/1) afforded **42** as a colorless oil (151 mg, 85%). ^1^H NMR (400 MHz, CDCl_3_) δ 7.38–7.27 (m, 11H), 5.41 (dd, *J* = 10.0, 9.1 Hz, 1H), 4.81–4.74 (m, 2H), 4.69 (d, *J* = 12.4 Hz, 1H), 4.58 (d, *J* = 12.2 Hz, 1H), 4.45 (d, *J* = 12.2 Hz, 1H), 4.23 (dd, *J* = 12.1, 5.6 Hz, 1H), 4.13 (dd, *J* = 12.1, 2.5 Hz, 1H), 3.85–3.75 (m, 3H), 3.33 (s, 3H), 2.09 (s, 3H), 2.01 (s, 3H). ^13^C NMR (100 MHz, CDCl_3_) δ 170.91, 169.75, 138.17, 138.12, 128.36, 128.32, 127.86, 127.65, 127.63, 127.41, 99.44, 73.96, 72.86, 71.86, 68.97, 68.08, 63.06, 55.00, 29.71, 20.92, 20.84.

*Methyl 2,3-di-O-benzyl-4,6-di-O-acetyl-β-D-glucopyranoside* (**43**) [50]. Following general process C, starting from **40** (90 mg, 0.19 mmol), purification by column chromatography (petroleum ether/ethyl acetate 3/1) afforded **43** as a colorless oil (78 mg, 88%). ^1^H NMR (400 MHz, CDCl_3_) δ 7.43–7.14 (m, 10H), 5.04 (dd, *J* = 10.0, 9.3 Hz, 1H), 4.91 (d, *J* = 10.9 Hz, 1H), 4.83 (d, *J* = 11.5 Hz, 1H), 4.70 (d, *J* = 11.0 Hz, 1H), 4.61 (d, *J* = 11.7 Hz, 1H), 4.34 (d, *J* = 7.7 Hz, 1H), 4.25 (dd, *J* = 12.2, 5.0 Hz, 1H), 4.08 (dd, *J* = 12.2, 2.4 Hz, 1H), 3.64– 3.52 (m, 5H), 3.49 (dd, *J* = 9.2, 7.7 Hz, 1H), 2.08 (s, 3H), 1.92 (s, 3H). ^13^C NMR (100 MHz, CDCl_3_) δ 170.85, 169.58, 138.25, 138.23, 128.40, 128.38, 128.15, 127.83, 127.78, 127.69, 104.72, 81.96, 81.52, 75.15, 74.91, 71.81, 69.72, 62.40, 57.29, 20.80.

*Methyl 2,3-di-O-benzyl-6-acetyl-α-D-glucopyranoside* (**44**) [39]. Following general process D, starting from **1a** (100 mg, 0.22 mmol), purification by column chromatography (petroleum ether/ethyl acetate 3/1) afforded **44** as a colorless oil (63.5 mg, 70%). ^1^H NMR (400 MHz, CDCl_3_) δ 7.40–7.27 (m, 10H), 5.00 (d, *J* = 11.3 Hz, 1H), 4.82–4.72 (m, 2H), 4.66 (d, *J* = 12.1 Hz, 1H), 4.62 (d, *J* = 3.5 Hz, 1H), 4.42 (dd, *J* = 12.1, 4.7 Hz, 1H), 4.21 (dd, *J* = 12.1, 2.2 Hz, 1H), 3.79 (t, *J* = 9.2 Hz, 1H), 3.74 (ddd, *J* = 10.0, 4.7, 2.2 Hz, 1H), 3.51 (dd, *J* = 9.6, 3.6 Hz, 1H), 3.42 (dd, *J* = 10.0, 8.9 Hz, 1H), 3.38 (s, 3H), 2.51 (s, 1H), 2.08 (s, 3H).

*Methyl 2,3-di-O-benzyl-6-acetyl-α-D-galactopyranoside* (**45**) [51]. Following general process D, starting from **15** (155 mg, 0.33 mmol), purification by column chromatography (petroleum ether/ethyl acetate 3/1) afforded **45** as a colorless oil (89 mg, 64%). ^1^H NMR (400 MHz, CDCl_3_) δ 7.41–7.22 (m, 10H), 4.81 (dd, *J* = 11.8, 2.2 Hz, 2H), 4.74–4.62 (m, 3H), 4.34–4.19 (m, 2H), 3.97 (t, *J* = 2.8 Hz, 1H), 3.96–3.82 (m, 3H), 3.37 (s, 3H), 2.55 (t, *J* = 1.5 Hz, 1H), 2.07 (s, 3H). ^1^H NMR (400 MHz, DMSO-*d*_6_) δ 7.43–7.11 (m, 10H), 4.98 (d, *J* = 5.0 Hz, 1H), 4.79 (d, *J* = 3.6 Hz, 1H), 4.73–4.55 (m, 4H), 4.20–4.06 (m, 2H), 4.06–4.00 (m, 1H), 3.77–3.74 (m, 2H), 3.67 (dd, *J* = 10.1, 3.1 Hz, 1H), 3.26 (s, 3H), 2.05 (s, 3H).

*Methyl 2,3-di-O-benzyl-6-acetyl-α-D-mannopyranoside* (**46**) [39]. Following general process D, starting from **39** (116 mg, 0.25 mmol), purification by column chromatography (petroleum ether/ethyl acetate 3/1) afforded **46** as a colorless oil (72 mg, 69%). ^1^H NMR (400 MHz, CDCl_3_) δ 7.42–7.23 (m, 10H), 4.78 (d, *J* = 1.7 Hz, 1H), 4.66 (q, *J* = 12.3 Hz, 2H), 4.52 (dd, *J* = 11.7, 36 Hz, 2H), 4.38 (qd, *J* = 12.0, 3.8 Hz, 2H), 3.95 (td, *J* = 9.7, 2.2 Hz, 1H), 3.79 (dd, *J* = 3.1, 1.8 Hz, 1H), 3.76–3.65 (m, 2H), 3.34 (s, 3H), 2.62 (d, *J* = 2.7 Hz, 1H), 2.09 (s, 3H).

*Methyl 2,3-di-O-benzyl-6-acetyl-β-D-glucopyranoside* (**47**) [51]. Following general process D, starting from **40** (190 mg, 0.41 mmol), purification by column chromatography (petroleum ether/ethyl acetate 3/1) afforded **47** as a white solid (111 mg, 65%). ^1^H NMR (400 MHz, CDCl_3_) δ 7.52–7.13 (m, 10H), 4.93 (dd, *J* = 11.2, 6.6 Hz, 2H), 4.72 (dd, *J* = 11.2, 5.8 Hz, 2H), 4.41 (dd, *J* = 12.1, 4.4 Hz, 1H), 4.36–4.25 (m, 2H), 3.57 (s, 3H), 3.49–3.34 (m, 4H), 2.65 (d, *J* = 2.2 Hz, 1H), 2.09 (s, 3H).

## Data Availability

Not applicable.

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
