# Peer review of "SnCl4 Promoted Efficient Cleavage of Acetal/Ketal Groups with the Assistance of Water in CH2Cl2"

_molecules, 2022, doi:10.3390/molecules27238258_

Round 1

Reviewer 1 Report

The manuscript titled " SnCl4-Promoted Efficient Cleavage of Acetal/Ketal Groups 2 with the Assistance of Water in CH2Cl2" demonstrates an acetal deprotection method. The protocol shows a broad substrate scope.

  however the following additions are required before publication.   1) how is this method superior to the previously published SnCl2 method. 2) Table 2, heading should be changed to deacetalation. 3) as the control experiment shows the formation of the product in the presence of HCl, it may be recommended to perform the experiment in degassed and distilled Dichloromethane. 4) as the SnCl4 is known to violently react with H2O to form tin oxide and HCl, it is not very likely to undergo an equilibrium between intermediate II and Intermediate III in figure 1a. 5) overall spell check and English correction are needed.

Author Response

Thanks a lot for all the constructive comments!

1) how is this method superior to the previously published SnCl2 method.

Answer: the published SnCl2 method requires long reaction time (12 h, entry 5 in table 1).

2) Table 2, heading should be changed to deacetalation.

Answer: we have revised it in light of the comment.

3) as the control experiment shows the formation of the product in the presence of HCl, it may be recommended to perform the experiment in degassed and distilled Dichloromethane.

Answer: we have performed an NMR experiment in dry d-chloroform. Benzaldehyde was observed instead of dicholorotoluene from the spectrum (Figure S1 in SI). The reaction was terminated when the trace amounts of water in the d-choloroform was consumed. These facts indicated that trace amounts of water play an indispensable role in the cleavage of the benzylidene group.  

4) as the SnCl4 is known to violently react with H2O to form tin oxide and HCl, it is not very likely to undergo an equilibrium between intermediate II and Intermediate III in figure 1a.

Answer: the above NMR experiment does not support the equilibrium mechanism. We have proposed a new mechanism in Figure 1b. 

5) overall spell check and English correction are needed.

Answer: overall spell check and English correction have been performed.

Reviewer 2 Report

See attached file.

Author Response

Thanks a lot for all constructive comments!

1) The correctness of the English language can be improved by thorough reading, thus eliminating the typos, incorrect word orders and sentences. E.g.:
page 1, row 12 and afterwards: “treatment of SnCl4 with various carbohydrates” should be replaced by “treatment of various carbohydrate with SnCl4”;
page 4, row 89 and afterwards: “Table 2” should be “Table 3”
page 5, rows 104-105: “5,6-isopropylidene” and “1,2-isopropylidene” should be used instead of “5,6-diisopropylidene” and “1,2-diisopropylidene”
page 6, row 151: “Based” should be written instead of “Base”;
page 6, row 157: “over” should be replaced by “from”

Abbreviations should be used after the first appearance of the original form and should not be used in the Abstract. This also applies to well-known abbreviations (TFA, DCM, etc.); it would be more stylish.

Answer: we have revised the manuscript in light of these comments.

2) In Table 2, reaction conditions should be given in separate columns. In this form, it is hard to get the necessary information from the table.

Answer: we have revised Table 2 in light of the comment.

3) Figure 1a) shows the proposed mechanism for the acetal cleavage with SnCl4. It can be seen that the byproduct is alpha,alpha-dichlorotoluene. Was this compound identified in any of the experiments? If not, it should be tried (e.g. with IR, NMR or mass spectroscopy).

Answer: we have performed an NMR experiment in dry d-chloroform. Benzaldehyde was observed instead of dicholorotoluene from the spectrum (Figure S1 in SI). The reaction was terminated when the trace amounts of water in the d-choloroform was consumed. These facts indicated that trace amounts of water play an indispensable role in the cleavage of the benzylidene group. The NMR experiment does not support the equilibrium mechanism. Thus we have proposed a new mechanism in Figure 1b.

4) In the case of compounds 6, 8, 12, 14, 20, 24, 28 and 30, migration of the acyl group(s) is(are) possible. Are these derivatives stable after storage? At least one case should be investigated.

Answer: the acyl migration usually occured under acid/base conditions. Thus the purified these compounds are enoough stable for storage. For example, the compound 6 remained stable after 3 years of storage at room temperature.